# IFN-γ +874 T/A Is Associated with High Levels of Sera CPK in Patients with Inflammatory Myopathies

**DOI:** 10.3390/cimb47070492

**Published:** 2025-06-27

**Authors:** Mónica Vázquez-Del Mercado, Beatriz Teresita Martín-Márquez, Erika Aurora Martínez-García, Marcelo Heron Petri

**Affiliations:** 1Departamento de Biología Molecular y Genómica, Instituto de Investigación en Reumatología y del Sistema Músculo Esquelético, Centro Universitario de Ciencias de la Salud, Universidad de Guadalajara, Guadalajara C.P. 44340, Jalisco, Mexico; dravme@hotmail.com (M.V.-D.M.); beatriz.martin@academicos.udg.mx (B.T.M.-M.); erikaaurora21@hotmail.com (E.A.M.-G.); 2División de Medicina Interna, Servicio de Reumatología SNP 004086 de la SECIHTI, Nuevo Hospital Civil Dr. Juan I. Menchaca, Guadalajara C.P. 44340, Jalisco, Mexico; 3Departamento de Fisiología, Centro Universitario de Ciencias de la Salud, Universidad de Guadalajara, Sierra Mojada No 950, Colonia Independencia, Guadalajara C.P. 44340, Jalisco, Mexico; 4Department of Medicine Solna, Karolinska Institutet, 17176 Stockholm, Sweden; 5School of Medical Sciences, Örebro University, 70182 Örebro, Sweden

**Keywords:** dermatomyositis, polymyositis, idiopathic inflammatory myopathies, polymorphism

## Abstract

**Aim of the study:** Idiopathic inflammatory myopathies (IIM) are autoimmune diseases with a low prevalence and incidence worldwide. The levels of IFN-γ production by T-lymphocytes are related to disease activity. IFN-γ +874 T/A (rs2430561) has been shown to alter the serum levels of IFN-γ in different pathologies. The aim of this work is to explore the role of IFN-γ +874 T/A polymorphism in IIM. **Methods:** Using a specific sequence primer-polymerase chain reaction (SSP-PCR), the genotype was defined for normal healthy controls (HC) and patients with IIM. Markers of muscle damage, clinical features and treatment were collected from chart at the time of diagnosis and at recruitment point. All the data were analyzed by demographic characteristics, genotype, type of IIM, treatment, clinical features, and enzymatic levels. **Results:** No association was found comparing the genotypes or alleles of the IIM patients vs. HC. On the other hand, the TT genotype, previously described as a high producer of INF γ, showed higher levels of CPK at diagnosis in IIM patients, whereas females at diagnosis and males in remission presented higher levels. **Conclusions:** Even with a limited number of patients due to the rarity of this disease, no association was found between the disease development. Further, the TT genotype promoted muscle damage due to CPK elevation in the sera compared to the TA/AA genotype in patients with IIM. This could be genetic evidence of the impact of IFN-γ in the disease activity of IIM patients. A larger cohort is needed to confirm these results.

## 1. Introduction

Idiopathic inflammatory myopathies (IIM) are autoimmune diseases with a low prevalence and incidence worldwide. This is why IIM are considered to be rare or orphan diseases. IIM are characterized by proximal muscle weakness, enzyme elevation, and the infiltration of both CD4^+^ and CD8 T^+^ lymphocytes in the inflamed muscle [1]. According to the evolving classification criteria, since first description by Bohan and Peter in 1975 [2], the clinical phenotypes recognized nowadays in the adult population according to European League Against Rheumatism and the American College of Rheumatology (EULAR/ACR) 2017 [3] and the European Neuromuscular Centre (ENMC) [4,5] are as follows: dermatomyositis (DM), polymyositis (PM), inclusion body myositis (IBM), immune mediated necrotizing myopathy (IMNM), overlap myositis (OV), cancer associated myositis (CAM), and anti-synthetase syndrome (ASS). Despite the increasing number of immune mediators described in IIM pathogeny [6,7], one of the main cytokines are the group responsible for the IFN signature [8,9]: type I and type II interferon (IFN) [10]. However, there is evidence that the levels of IFN-α in the sera do not correlate with disease activity [11].

The main cytokines produced by T-lymphocytes infiltrated in the muscle of patients with IIM is usually type II interferon, more specifically, IFN-γ. Both T-lymphocyte and perforin are highly present among muscle fibers in muscle biopsies [12], suggesting that those T-lymphocytes are not only promoting inflammation in the tissue, but also are activated by an antigen that still remains to be identified. Experimental evidence indicates that interferon-gamma may impair skeletal muscle regeneration by inhibiting both myoblast proliferation and creatine kinase activity, suggesting a potential mechanistic link between IFN-γ signaling and the reduced muscle repair capacity observed in inflammatory myopathies [13]. It is known that the levels of IFN produced by T-lymphocytes are related to the disease activity [14], and high levels of IFN-γ in the blood are associated with DM and PM [15]. Interesting, high levels of IFN-γ have been reported with activity of the disease in patients that are resistant to intravenous immunoglobulin treatment [16]. On the other hand, there is evidence of patients with active IIM that present low levels of IFN-γ and CD3-positive cells in the peripheral blood, along with a higher number of B-lymphocytes and increased IL-4 compared with non-active patients [17]. A mouse model of Duchene disease showed that compared with IFN-γ knockout mice, IFN-γ^+/+^ had more muscle damage with the suppression of M2 macrophages [18], confirming the role of IFN-γ with a more aggressive disease, muscle damage, and more inflammation characterized by M1 macrophages and T-lymphocyte infiltrated in the muscle.

Among the different single nucleotide polymorphisms (SNP) related to IFN-γ, IFN-γ +874 T/A (rs2430561) is related to not only systemic lupus erythematosus [19,20], Grave’s disease [21,22] and thyroiditis of Hashimoto [22], but it has also been proven to be associated with the A allele, producing lower levels of IFN-γ at a protein level compared with those that have the T allele. The polymorphism is recognized by the transcription factor nuclear factor-κ-B (NF-κ-B) and it changes its activity in the promoter region. At the protein level, the IFN-γ +874 T alleles are associated with a high IFN-γ expression in vivo [22,23]. IFN-γ +874 T/A polymorphism has not only been associated with autoimmunity, but also with cancer [24,25], infection [23,26,27], and even aging [28].

As the alleles are already known to change the protein expression of IFN-γ, the goal of this study was to describe the influence of the IFN-γ +874 T/A polymorphism in patients with IIM, as well as their clinical features, in a group of 48 patients recruited from Mexico City and Guadalajara.

## 2. Material and Methods

### 2.1. Patients

Subjects for the study were recruited from rheumatology outpatient clinics at Guadalajara and Mexico City. The protocol was approved by the Institutional Review Board (IRB) committee of Hospital Civil de Guadalajara, “Dr. Juan I. Menchaca”, under the register 969/10 on 12 February 2010. Ethical approval was obtained before enrolment; all subjects signed written consent and were informed about the study according to the Declaration of Helsinki. Forty-eight patients with IIM (12 PM and 36 DM) fulfilled the Bohan and Peter criteria [2]. The clinical features were recorded at the time of recruitment. As a control group, we included 75 healthy subjects paired by sex and age.

### 2.2. Laboratory Studies

The blood was drawn by peripheral venipuncture for general studies including blood cell count, glucose, and enzymes, such as creatine phosphokinase (CPK), lactic dehydrogenase (LDH), aspartate aminotransferase (AST), and alanine aminotransferase (ALT), were recorded from the chart at the time of diagnosis (referred as Starting) and at the time of recruitment (referred to as Current).

### 2.3. Genotyping

Blood was obtained with EDTA and the DNA was extracted according to Miller modified technique [29]. The DNA was kept at −20 °C until the PCR analysis. The genotypes were analyzed using specific sequence primer-polymerase chain reaction (SSP-PCR), as described elsewhere [22]. Using the following 5′-TCAACAAAGCTGATACTCCA-3′ as a common reverse primer and in separated PCRs, the specific primers were used as forward primers 5′-TTCTTACAACACAAAATCAAATCA-3′ for the A allele and 5′-TTCTTACAACACAAAATCAAATCT-3′ for the T allele to obtain a 261 pb fragment. To ensure that the PCR reaction worked, the 426 bp band from the Growth Hormone was amplified using the following primers: Forward 5′-GCCTTCCAACCATTCCCTTA-3′ and reverse 5′-TCACGGATTTCTGTTGTGTTTC-3′. The conditions for the PCR reaction were 10 cycles (94 °C for 3 min, 94 °C for 30 s, 62 °C for 30 s, and 72 °C for 50 s), followed by 25 cycles (94 °C for 30 s, 56 °C for 30 s, and 72 °C for 50 s). The bands were confirmed with 2% agarose gel stained with GelRed™ (Biotium, Fremont, CA, USA).

### 2.4. Statistical Analysis

Hardy–Weinberg equilibrium was computed in the controls. Fisher exact test was used to compare the genotype and allele frequencies and clinical features between IIM patients and healthy subjects. Two-way ANOVA with Holm–Sidak as the post hoc test was used to compare the levels of muscle enzymes among the genotypes, type of myopathy, and sex; then, the Mann–Whitney test was used to compare the presence/absence of the polymorphic allele in patients with IIM. A *p* value of <0.05 was considered significant and the results are represented by mean ± SEM (standard error of the mean). All of the analyses and graphics were performed using GraphPad^®^ Prism v3.5 (Boston, MA, USA).

## 3. Results

### 3.1. Demographic Distribution and Time with the Disease

For the controls, 31 males and 44 females were recruited with a mean age of 36 ± 5 years of age. The patients with IIM were 10 males and 26 females, with DM of 42 ± 6 years old and 7 males and 5 females with PM with 47 ± 4 years old. There was no difference in age distribution among the patients with DM, PM, or healthy controls.

The time from diagnosis (starting) and time to study recruitment (current) were registered, and time with disease in months was calculated and divided by sex and type of IIM. Independent of the type of IIM, female patients presented, in general, a longer evolution time compared to males, although it was not significant. More details based on the type of IIM are described in Table 1.

### 3.2. Prevalence of the Genotype and Alleles

The healthy controls were calculated using Hardy–Weinberg Equilibrium (χ^2^ = 0.123 and *p* value = 0.725). The genotype distribution among the healthy subjects was TT = 39 (52%), TA = 31 (41.3%) and AA = 5 (6.7%), with an allele distribution of T = 109 (73%) and A = 41 (27%). In the IIM group, the genotype TT = 32 (66.6%), TA = 15 (31.3%), and AA = 1 (2.1%). There were no associations with genotype or allele distributions, among the healthy subjects and IIM patients. A sub-analysis was performed with the type of IIM, and showed no association. The discrimination of the genotypes and alleles by type of IIM are in Table 2.

### 3.3. Enzymes Levels and Clinical Features in DM/PM

Since only one patient presented the AA genotype in the disease groups, the analysis of all the enzymes levels was done to compare TT vs. TA/AA to understand the impact of A allele in the disease. The enzymes collected at the recruitment point had no significant difference between the genotypes on the other hand, for the diagnostic points referred to as “starting”, there was no significance in LDH, AST, and ALT, but for the CPK levels, the patients carrying the TT genotype presented higher levels compared with TA/AA patients (Figure 1A). The clinical features, such as Gottron’s papules, Shawl sign, heliotrope, fever, dyspnea, and weight lost, presented no differences between the different genotypes analyzed.

### 3.4. Enzymes Levels by the Type of IIM

The TT and TA/AA genotypes were analyzed by type of IIM as the independent groups. Among the patients with DM, the patients with the TT genotype present higher levels of starting CPK (4346.2 ± 957.9 U/L) compared with TA/AA (1068.1 ± 493.5 U/L *p* < 0.05) (Figure 2A). For the patients with PM, no differences in CPK were observed, but current ALT was higher with the TT genotype (75.1 ± 2.7 U/L) compared to TA (20.2 ± 1.9 U/L *p* < 0.05) (Figure 2D), whereas no differences were observed in DM.

### 3.5. Enzymes Levels and Clinical Features in IIM by Sex

Once the groups were classified by gender, no differences were found for clinical features or levels of LDH, AST, and ALT. On the other hand, the CPK level for both starting and recruitment (current) time points presented different behaviors based on sex. The starting CPK levels presented no statistical difference between genotype for male patients, but female patients with the TT genotype presented higher CPK levels compared with the TA/AA genotype (Figure 3A). For the patients in remission, a different was found when exploring the impact of sex with the genotype. The male patients with the TT genotype presented higher levels of CPK compared with TA/AA genotype (Figure 3B).

## 4. Discussion

IIM is a rare autoimmune disease that is predominantly present in females, characterized by proximal muscle weakness; enzyme elevation of CPK, LDH, AST, and ALT in sera; and by muscle fibers infiltrated with lymphocytes in biopsies [1]. The lymphocytes in the muscle are predominantly T-lymphocytes CD4^+^ and CD8^+^, although it is known that DM is predominantly a Th2 response with B-lymphocyte involvement based on the presence of autoimmune antibodies that often presents with specific clinical features. The role of activation of the classic way of complement is now recognized by some authors as a vasculopathy linked in most of the cases to the positivity of anti-melanoma gene differentiation factor 5 (anti-MDA5) [9]. Meanwhile, PM is mainly characterized by a Th1 response and is quite frequent the absence of circulating autoantibodies [15]. This is because T-lymphocytes are the main source of IFN-γ and the T allele of the polymorphism IFN-γ +874 is associated with higher levels of IFN-γ in vivo [22,23]. The opposite effect is also true for the A allele, which has been associated with a lower expression of IFN-γ in the sera from cancer patients [25] and other autoimmune diseases [21].

The present study shows that IFN-γ +874 T/A is not associated with IIM, but the TT genotype could impact CPK levels. Also, a sex-associated pattern was observed, whereas the starting CPK levels at diagnosis were higher in females with the TT genotype, whereas the same genotype presented higher CPK levels at recruitment point in males. It is not possible to exclude that the time with the disease and/or treatment could play a role in this observation. Females presented a longer evolution time compared to males. Due to the low number of patients and because the disease is rare, we can only speculate that it could be due to its less aggressive nature or the fact that females seek more medical attention compared to males in Mexican populations.

Because this is a rare autoimmune disease, patients from different cities were recruited. Even with the limited number of patients, the results showed that in general, patients with the T allele, which was previously associated with higher levels of IFN-γ, also present with higher enzymatic levels of CPK in different time points and genders. It is crucial to clarify that at the recruitment point as overall, there was no difference between the genotypes. This could be because of treatment with glucocorticoids. When analyzed by type of IIM, the result showed that DM patients carrying the A allele had lower levels of CPK in the sera, confirming the findings found in the IIM group. On the other hand, only Current ALT in PM patients showed no significant differences in clinical features.

It is known that in autoimmune rheumatic diseases, women are predominantly affected. Another strong point of this work is that focusing on CPK grouped by sex showed no difference among the male patients, but analyzing the female patients with the TT genotype had around 13-fold more CPK than the women that carried the A allele presented a more aggressive disease or implied resistance to treatment. To the best of our knowledge, this is the first work that relates the TT genotype IFN-γ +874 SNP with CPK elevation, particularly IIM. How the allele T, present on the T-lymphocyte infiltrated on the muscle of the patients with IIM, induces the CPK elevation in IIM patients is unknown, but we were able to observe that the levels of CPK present in these patients presented a more aggressive phenotype, leading to more muscle damage and increasing CPK levels for these patients. At this point, it is important to remember two important facts: (1) the classical DM, positive for the anti-Mi2 antibody is characterized by sometimes extremely high CPK levels at the diagnosis time, but with a very good clinical prognosis [30]. (2) CPK might remain elevated, no more than two-fold from the maximum reference level but this finding is not related to IIM activity but instead to a lack of rehabilitation therapy, [31]. Although no IFN-γ levels were missing for these patients at different time points, the evidence that the T allele of IFN-γ +874 is associated with higher levels of IFN-γ is robust. This has also been explored in other contexts, such as sepsis and infection. The current data agreed that the TT genotype was associated with higher levels of IFN-γ and sepsis susceptibility [32], whereas in COVID 19 infection, the A allele was associated with more critical patients while the T allele was protective and more present in healthy controls [27,33]. Further, higher levels of IFN-γ have been reported in other autoimmune diseases, such as systemic lupus erythematosus [19] and chronic fatigue syndrome [34], as well as in a mouse model of Duchene syndrome, where the high levels of IFN-γ promoted muscle damage [18].

## 5. Conclusions

Taken together, the data could show that IFN-γ +874 T/A polymorphism is not associated with a risk of developing IIM. Such a result could be related to the small sample size due to the rarity of the disease, However, the T allele, previously associated with higher cytokine production, also promotes muscle damage with CPK elevation in the sera, leading to a risk of worse muscle inflammation in IIM at the diagnosis time, mainly if the patient was a female bearing the TT genotype. Whereas the male patients with TT genotype presented higher levels of CPK at recruitment time compared with TA/AA genotype. A larger study and different ethnic populations should be considered to confirm these findings and better understand the role of the IFN-γ +874 T/A polymorphism in the pathogenesis of IIM.

## Figures and Tables

**Figure 1 cimb-47-00492-f001:**
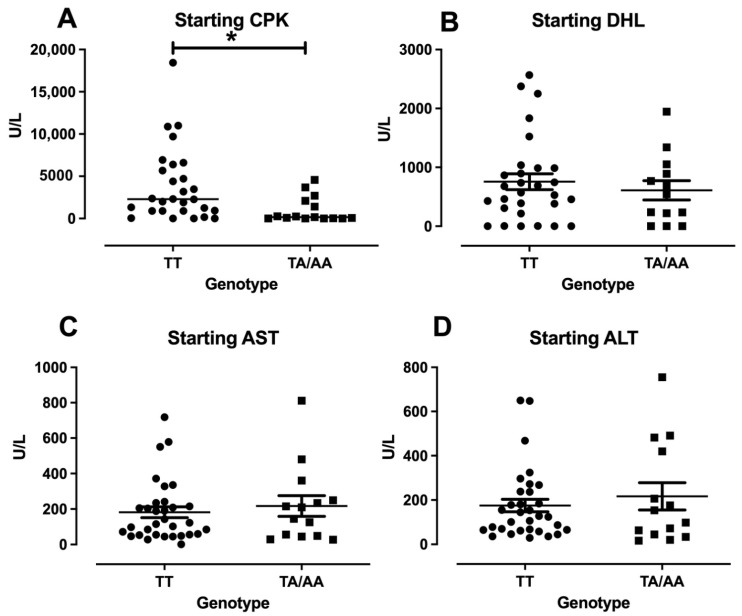
Enzyme levels at the diagnostic point (Starting) of CPK, DHL, AST, and ALT in IIM. The graphics represent the different enzymes in the sera of patients with IIM at the diagnosis grouped by genotype. The CPK levels (**A**), DHL (**B**), AST (**C**), and ALT (**D**). The * represent a *p*-values < 0.05.

**Figure 2 cimb-47-00492-f002:**
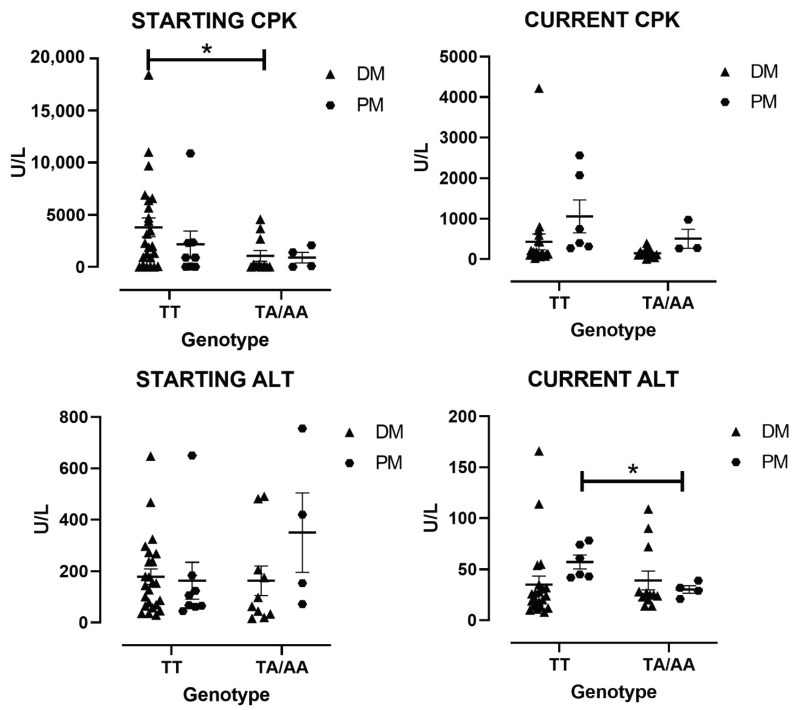
Enzyme levels at the diagnostic point (Starting) and current CPK and ALT by type of IIM grouped by genotype. The graphics represent the different enzymes in sera of patients by type of IIM by genotype. Starting CPK levels (**A**), current CPK levels (**B**), starting ALT levels (**C**), and current ALT levels (**D**). The * represent a *p*-values < 0.05 DM, dermatomyositis; PM, polymyositis.

**Figure 3 cimb-47-00492-f003:**
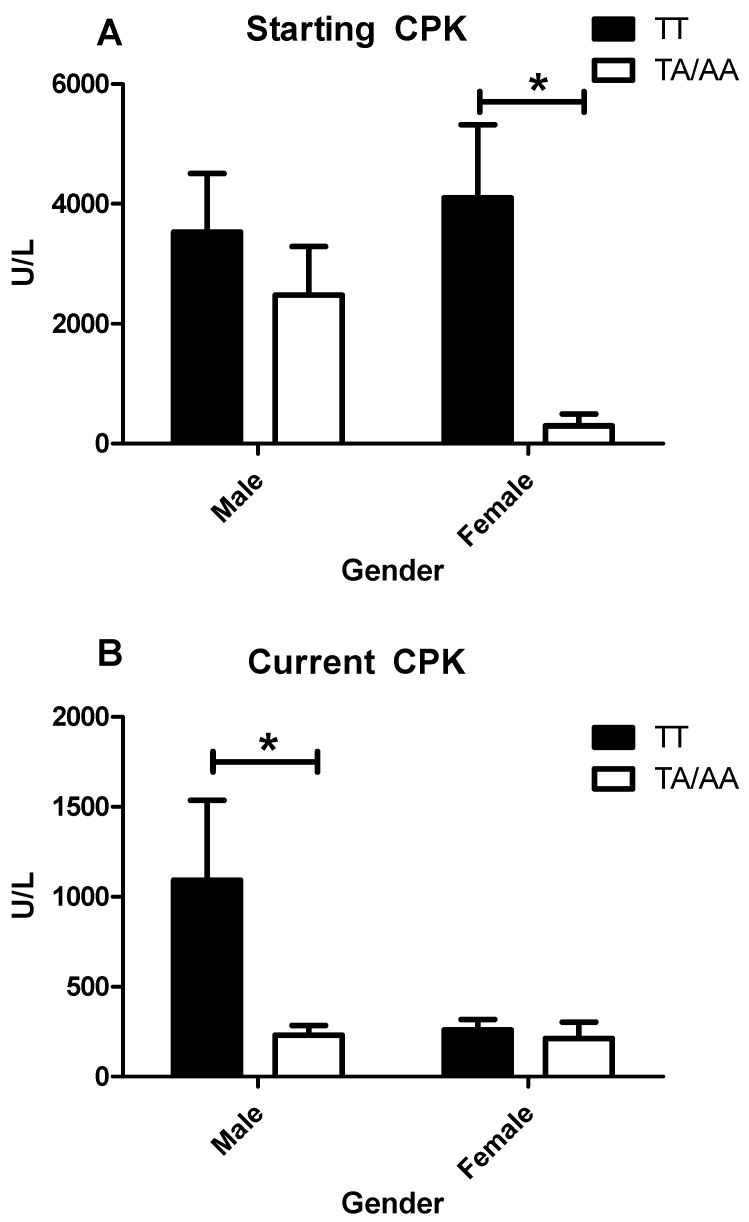
CPK levels at different time points grouped by sex and genotype. The graphic represents the levels of CPK at starting point (diagnosis) of male and female patients with IIM grouped by genotype (**A**) and the current levels of CPK (recruitment point) for male and female patients with IIM grouped by genotype (**B**). The * represent a *p*-values < 0.05.

**Table 1 cimb-47-00492-t001:** Geographic distribution for sex and age of idiopathic inflammatory myopathy (IIM) patients with time with disease and healthy controls (HC). DM, dermatomyositis; PM, polymyositis; SEM, standard error of the mean.

Geographic Distribution of IIM Patients and HC					
	Male	Time with Disease (Months) Mean ± SEM	Female	Time with Disease (Months) Mean ± SEM	Age Mean ± SEM
DM	10	3.7 ± 0.3	26	17.2 ± 1.3	42 ± 6
PM	7	2.54 ± 0.5	5	23.2 ± 9.5	47 ± 4
HC	31		44		36 ± 5

**Table 2 cimb-47-00492-t002:** Genotypes and frequencies of IFN-γ +874 polymorphism in idiopathic inflammatory myopathy (IIM) for patients and healthy controls (HC). DM, dermatomyositis; PM, polymyositis; NS, not significant.

Genotype and Frequencies of IFN-γ +874 Polymorphism in IIM Patients and HC					
Genotype	DM/PM	DM	PM	HC	
	N = 48 (%)	N = 36 (%)	N = 12(%)	N = 75 (%)	*p* Value
TT	32 (66.6)	24 (66.8)	8 (66.7)	39 (52)	NS
TA	15 (31.3)	11 (30.5)	4 (33.3)	31 (41.3)	NS
AA	1 (2.1)	1 (2.7)	0 (0)	5 (6.7)	NS
Allele					
T	79 (82)	60 (81)	20 (83)	109 (73)	NS
A	17 (18)	14 (19)	4 (17)	41 (27)	NS

## Data Availability

The data presented in this study are available on request from the corresponding author.

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
