# Peer review of "IFN-γ +874 T/A Is Associated with High Levels of Sera CPK in Patients with Inflammatory Myopathies"

_cimb, 2025, doi:10.3390/cimb47070492_

Round 1
Reviewer 1 Report
Comments and Suggestions for Authors
The manuscript by Mercado et al. explored the role of the genetic polymorphism, IFN-gamma +874 T/A, in Idiopathic Inflammatory Myopathies (IIM). They employed Specific Sequence Primer-PCR (SSP-PCR) to genotype 72 healthy control subjects and 49 IMM patients. They found no association between the genotype and the disease risk, but a higher level of creatine phosphokinase (CPK) in the TT genotype as compared to TA/AA genotypes. Thus, they concluded that IFN-gamma +874 T/A is associated to serum CPK level in IIM.
Comments and suggestions
- Line 91, "Forty-eight patients with IIM (12 PM and 37 DM)" Isn't it 12+37=49?
- Line 137, Table 1, it would be better to include sex in addition to age in the demographic information.
- Line 133, "In the IIM group, the genotype TT=32 (65.3%), TA=16 (32.6%) and AA= 1(2.1%)." The genotype AA is so infrequent in this study, thus the lack of association could be a power issue. The authors are recommended to acknowledge the limitation of small sample size.
- If I didn't read it wrong, Figure 1 and 2 look the same—data points are identical between the two figures. The authors should double check.
- The association between the TT allele and higher levels of serum CPK is only found in female patients at the disease onset, or male patients at the recruitment point. How to interpret this interaction between sex and disease progress. The authors are recommended to make more interpretation about Figure 3.
Author Response
Thank you kindly for the time and effort to improve our work. We as a group appreciate the comment and hope to be able to answer all the inquiries.
We have also changed some parts to improve the English as requested from the reviewer 2 and we hope that this would fulfill his wishes. If the scientific content is accepted for publication and English review is still required, we can discuss use the MDPI English reviewing services.
Now the manuscript has been revised after the reviewers’ comments as followed:
Reviewer 1
- Line 91, "Forty-eight patients with IIM (12 PM and 37 DM)" Isn't it 12+37=49?
Answer: Thank you for your comment. Indeed, it was an error in the table where we had 27 females with DM instead of 26. The table 1 was corrected, and since the text was done based on the table, the text was revised and corrected also. Same mistake was done with the healthy controls that the numbers did not match the text with the table. Now after revision, all the numbers are corrected.
- Line 137, Table 1, it would be better to include sex in addition to age in the demographic information.
Answer: The information is now both in a new topic in results and a revised table one.
- Line 133, "In the IIM group, the genotype TT=32 (65.3%), TA=16 (32.6%) and AA= 1(2.1%)." The genotype AA is so infrequent in this study; thus the lack of association could be a power issue. The authors are recommended to acknowledge the limitation of small sample size.
Answer: Dear Reviewer, you are right, the sample size is limited, it is clear on the conclusions on the abstract. It is important to point out that this is not due to resources, mainly due to the disease presentation that is rare, hance why we collected from two different regions in Mexico to compensate such limitation. It is acknowledged in the discussion on lines 188-190. Now with the revised document it was further reinforced in the conclusions (new tittle) on line 248. The lack of observation of the AA genotype is due to the population and we unfortunately can’t affect such prevalence. Interesting, such observation was also observed in the healthy control. Moreover, this observation is confirmed by the Hardy-Weinberg Equilibrium as exposed in the results. Therefore, no more comments were done in the discussion about the prevalence in AA in this study.
- If I didn't read it wrong, Figure 1 and 2 look the same—data points are identical between the two figures. The authors should double check.
Answer: Thank you for the comment. For some reason the figure 2 was placed twice in the text, now the correct figure 1 is on the revised version.
- The association between the TT allele and higher levels of serum CPK is only found in female patients at the disease onset, or male patients at the recruitment point. How to interpret this interaction between sex and disease progress. The authors are recommended to make more interpretation about Figure 3.
Answer: Thank you for your comment. Such results were also comments from the other reviewer. We also added the time with disease which showed that females had more time with the disease compared to male patients. Such finding was also added on the text. Due to the design of the study, no mechanisms were explored, and we are currently working more insight about this finding. But we feel that such observation is important, although hard to explain, and was important to be included in this work.
Reviewer 2 Report
Comments and Suggestions for Authors
This paper evaluates a small group of patients with inflammatory myopathies and the role of a specific interferon polymorphism in terms of disease activity. There are major flaws with the design and discussion points of the study. There is no explanation as to why the specific outcome measures were chosen (CPK and ALT). The second set of measures are being referred to as "Current" however there are no details on the timing of data collection and the details of treatments and clinical status of these patients during this time. Therefore it is essentially not possible to make reliable inferences from the changes in these measures, in relation to the genotypic and sex differences of the patients as claimed. Last but not least, there are a few mistakes related to the use of English throughout the manuscript, which impede the flow and understanding.
Author Response
Thank you kindly for the time and effort to improve our work. We as a group appreciate the comment and hope to be able to answer all the inquiries.
We have also changed some parts to improve the English as requested from the reviewer 2 and we hope that this would fulfill his wishes. If the scientific content is accepted for publication and English review is still required, we can discuss use the MDPI English reviewing services.
Now the manuscript has been revised after the reviewers’ comments as followed:
Reviewers 2
- This paper evaluates a small group of patients with inflammatory myopathies and the role of a specific interferon polymorphism in terms of disease activity. There are major flaws with the design and discussion points of the study. There is no explanation as to why the specific outcome measures were chosen (CPK and ALT).
Answer: Thank you for the comment. The study did not focus on the mechanisms related to the polymorphism; therefore it is hard to be able to explain such clinical effects. Since the different enzymes associated with the muscle damage are also used for diagnosis and following up of the patients, we used these enzymes in this study. We explored the polymorphism and found (although with limited sample size) the association with such enzymes. The comment on the evolution time done by you were very insightful and we found difference in evolution time in this population based in sex, which could be also an co-found. This is mainly an observation that this work adds to the field and due to study design, we can not draw a conclusion of a direct mechanism. Although all the associations with higher enzymatic levels were observed in the TT genotype. We are currently exploring the possibility to further explain such mechanism but far away to be able to add to the current work.
- The second set of measures are being referred to as "Current" however there are no details on the timing of data collection and the details of treatments and clinical status of these patients during this time. Therefore it is essentially not possible to make reliable inferences from the changes in these measures, in relation to the genotypic and sex differences of the patients as claimed.
Answer: Thank you for your comment, it was an excellent comment that we as authors had not explored. With the revised manuscript, now the Table 1 and the results includes the time with the disease from the diagnostic date to the recruitment date in months. Interesting was male presented shorter time from the diagnosis to recruitment compared to women (not significant statistically) it was also added on the discussion. It is unclear to us now if this evolution time could explain the CPK levels observed with genotype and sex. Since the same genotype was associated with higher PCK levels at different time points. Since the males had lower evolutions time, maybe the treatment were not able to reduce the inflammation and muscle damage leading to higher levels. Such reasoning is also in the discussion on the revised manuscript.
- Last but not least, there are a few mistakes related to the use of English throughout the manuscript, which impede the flow and understanding.
Answer: Thank you for the comment, Since the other reviewer did not mention the lack of English we will discuss the possibility to use the service provided by MDPI to address such issue if you are not please with the current version and the manuscript is accepted for publication.
Round 2
Reviewer 1 Report
Comments and Suggestions for Authors
The authors have revised the manuscript and addressed my concerns. I do not have more questions now.
Reviewer 2 Report
Comments and Suggestions for Authors
The authors made some improvements in the manuscripts based on the previously provided recommendations. However, there still remains major flaws with the study. Similar to previously stated, it is not possible to draw conclusions translatable to clinical practice based on the results of this study (ie. the clinical significance of the findings are not clear).